# Assessment of Active Video Games’ Energy Expenditure in Children with Overweight and Obesity and Differences by Gender

**DOI:** 10.3390/ijerph17186714

**Published:** 2020-09-15

**Authors:** Cristina Comeras-Chueca, Lorena Villalba-Heredia, Marcos Pérez-Llera, Gabriel Lozano-Berges, Jorge Marín-Puyalto, Germán Vicente-Rodríguez, Ángel Matute-Llorente, José A. Casajús, Alejandro González-Agüero

**Affiliations:** 1Faculty of Health and Sport Science (FCSD), Department of Physiatry and Nursing, Universidad de Zaragoza, 50009 Zaragoza, Spain; ccomeras@unizar.es (C.C.-C.); glozano@unizar.es (G.L.-B.); gervicen@unizar.es (G.V.-R.); amatute@unizar.es (Á.M.-L.); 2GENUD (Growth, Exercise, Nutrition and Development) Research Group, Department of Physiatry and Nursing, Universidad de Zaragoza, 50009 Zaragoza, Spain; lvillalbaheredia@unizar.es (L.V.-H.); marcospelle21@gmail.com (M.P.-L.); jmarinp@unizar.es (J.M.-P.); joseant@unizar.es (J.A.C.); 3EXERNET Red de Investigación en Ejercicio Físico y Salud para Poblaciones Especiales, Spain; 4Faculty of Health Science, Department of Physiatry and Nursing, Universidad de Zaragoza, 50009 Zaragoza, Spain

**Keywords:** active video games, energy expenditure, gender differences, children, obesity

## Abstract

(1) Background: Childhood obesity has become a main global health problem and active video games (AVG) could be used to increase energy expenditure. The aim of this study was to investigate the energy expenditure during an AVG intervention combined with exercise, differentiating by gender. (2) Methods: A total of 45 children with overweight or obesity (19 girls) performed an AVG intervention combined with exercise. The AVG used were the Xbox Kinect, Nintendo Wii, dance mats, BKOOL cycling simulator, and Nintendo Switch. The energy expenditure was estimated from the heart rate recorded during the sessions and the data from the individual maximal tests. (3) Results: The mean energy expenditure was 315.1 kilocalories in a one-hour session. Participants spent the most energy on BKOOL, followed by Ring Fit Adventures, Dance Mats, Xbox Kinect, and the Nintendo Wii, with significant differences between BKOOL and the Nintendo Wii. Significant differences between boys and girls were found, but were partially due to the difference in weight, VO_2max_, and fat-free mass. (4) Conclusions: The energy expenditure with AVG combined with multi-component exercise was 5.68 kcal/min in boys and 4.66 kcal/min in girls with overweight and obesity. AVG could be an effective strategy to increase energy expenditure in children and adolescents with overweight and obesity.

## 1. Introduction

Obesity has become a main global health problem because of the alarming increase in its prevalence [1]. The World Health Organization (WHO) even refers to obesity as a “global pandemic” [2,3]. Childhood obesity is considered an important public health issue of the 21st century [4,5], especially in developing countries but also in developed countries, with a prevalence of 18% of children and adolescents aged 5–19 with overweight or obesity in 2016 [6].

Overweight and obesity involve serious consequences, including an increase in the risk of developing cardiovascular and cardio-metabolic diseases such as type 2 diabetes, hypertension, and metabolic syndrome; and psychosocial problems such as low self-esteem, low self-confidence, low self-efficacy, low motivation for physical activity, bullying, or difficulties in establishing relationships [7,8]. From an economic point of view, childhood obesity represents an economic expense for the health-care system [9]. In addition, overweight and obese children are likely to remain obese during adulthood and will be likely to develop all types of cardiovascular and metabolic pathologies [4].

According to the WHO, insufficient physical activity is one of the main risk factors for death worldwide and a key risk factor for non-communicable diseases such as cardio-metabolic and cardiovascular diseases and cancer [10]. Children and adolescents should participate in at least 60 min of moderate to vigorous intensity physical activity daily, and could expect additional benefits with a greater amount of physical activity [11]. Nevertheless, a large proportion of the children and adolescents do not meet the public health recommendations of physical activity [12]. Over 80% of the world’s adolescent population is insufficiently physically active, with girls being less active than boys and inactivity rising with age [11,13], and the relationship between physical inactivity and obesity is well known [14].

The development of new technologies with the industrial revolution has led to a decline in physical activity and energy expenditure, but also to an increase in sedentary behavior among children and adolescents [13]. The American Academy of Paediatrics recommends not to exceed 2 h of screen time per day [15], but approximately a third of the European children included in IDEFICS study exceeded the recommendations [16]. Screen-based behaviors such as using smartphones, tablets, or computers; watching television; or playing videogames have become the preferred sedentary behaviors in free-time daily living among children and adolescents [17,18]. Gender differences have been shown for screen time, with boys spending more time in front of a screen and specifically playing video games [18].

The alarming increase in the prevalence of obesity and its dangerous consequences points out the need to confront it from childhood. It is well known that exercise is an effective tool to fight against obesity [19], but the main challenge is to ensure adherence to exercise in children with overweight and obesity [20]. Obesity prevention and treatment strategies have not been successful in the long term, and it is necessary to find new effective strategies to promote physical activity and to combat childhood obesity [10,18]. According to Ara et al. [21], interventions with children and adolescents with overweight or obesity are more likely to be effective and successful when working on reducing sedentary time and increasing moderate to vigorous physical activity rather than restricting dietary energy intake.

There is controversy about the influence of gender on energy expenditure. Drenowatz et al. [22] showed that the absolute and relative energy expenditure were higher in males compared with females. This could be due in part to the lower resting energy expenditure of females compared with males [23]. However, Geer et al. [24] reported similar resting energy expenditure in males and females when normalized to lean body mass. Other studies investigating differences in energy expenditure between males and females do not show significant differences. Klausen et al. [25] examined the effects of gender on energy expenditure independent of differences in body composition, showing that no significant differences between males and females for energy expenditure and basal metabolic rate were observed. These results were supported by Grund et al. [26], who concluded that gender had no significant effects on energy expenditure in prepubertal children. The results showed no gender differences in the resting energy expenditure, total energy expenditure, and activity-related energy expenditure, in addition to that resting energy expenditure and total energy expenditure were significantly related to fat-free mass [26].

Active-video games (AVG) have been proposed as a promising alternative, seeking to ally with new technologies rather than fighting against them [27,28]. AVG require body movement and the involvement of large muscle groups and, therefore, can be used to increase energy expenditure [27]. AVG interventions seems to be an effective strategy to promote the light-to-moderate physical activity among children and adolescents, being a good alternative to replace some sedentary behaviors [29,30]. AVG can be used to increase the energy expenditure to meet the daily recommendations of physical activity [31,32]. Some reviews show that AVG could contribute to promote a physically active lifestyle [32,33]. AVG interventions need to be structured and supervised. because no effects were found on physical activity or energy expenditure in home-based and/or unstructured AVG interventions [34,35].

Although scientific evidence seems to show the potential of AVG to increase energy expenditure and help children and adolescents achieve the daily physical activity recommendations to fight against obesity, the possibilities and effects of exercise sessions with AVG need to be investigated. It is unclear whether the intensity produced by AVG is enough to be considered a tool on which to base an intervention to prevent or treat obesity. In addition, it would be interesting to investigate the differences in energy expenditure when playing AVG between boys and girls. Gao et al. [33] pointed out the importance of taking gender differences into account, so an analysis of energy expenditure stratifying by gender is interesting.

A new AVG, the Ring Fit Adventures for Nintendo Switch, has been developed with the main objective of exercising. The Ring Fit Adventures is cutting-edge technology in AVG, and that is why it is necessary to study the potential of this AVG to fight against child obesity. What makes this AVG new and different is the use of a resistance ring to exert a resistance in the upper limb exercises and the use of a leg strap to control the movements of the lower extremities through a device with an accelerometer. The structure is like that of a conventional video game, with levels to overcome, with the difference that they are overcome with body movements and exercises. The intensity and difficulty of the exercises increases as the levels are surpassed. The detailed explanation of the exercises that are carried out, as well as a visual example, is noteworthy. In addition, it can be used by both adults and children.

Therefore, the aim of this study was to investigate the energy expenditure of combined sessions using several AVG, as well as to analyze the energy expenditure of each of the AVG used, especially the innovative AVG of Ring Fit Adventures. Furthermore, gender differences in energy expenditure during the sessions were examined.

## 2. Materials and Methods

This study was performed in accordance with the ethical guidelines of the Helsinki Declaration of 1964 (revised in Fortaleza, 2013), and were reviewed and approved by the Research Ethics Committee of the Government of Aragon (certificate numberº 11/2018, CEICA, Spain). All the participants and their parents or legal guardians were informed of the nature and possible risks of the experimental procedures before their written informed consent was obtained.

This is a cross-sectional study which is part of a larger randomized controlled trial which is registered in clinicaltrials.gov (identification number NCT04418713).

### 2.1. Participants

A total of 45 children with overweight or obesity met the inclusion criteria and participated in the study. The inclusion criteria were as follows: the participants had to be between the ages of 9 and 12 years, in Tanner I or II stage and not having had menarche, with overweight or obesity calculated by body mass index (BMI) and following the cut points of Cole et al. [36], without contraindications for the practice of physical exercise, and without pathologies that worsen with physical exercise. Tanner’s stage was evaluated by a medical doctor. In addition, the exclusion criteria for the participants’ daily lives were the following: participating in regular high-level or high-intensity extracurricular physical activities, following any special diet regime, and taking any medication that may interfere with the variables evaluated.

The participants were recruited from medical centers through their pediatricians or from schools of Zaragoza (Spain). The parents and pediatricians were informed about the development of the activity, the results, and the progress of the children.

### 2.2. Intervention

The participants were requested to attend to 3 sessions per week with a duration of 60 min each one. A total of 91 sessions were recorded. The sessions were composed of 10 min of warm-up, including joint mobility; dynamic flexibility; muscle activation; core, balance, and coordination exercises. This was followed by the main part, which consisted of 45 min of exercise with a combination of AVG and multicomponent exercise, following a circuit training dynamic where the participants were continuously rotating from AVG to exercises, and finally a 5 min cool-down part to lower the heart rate and end the session with static flexibility routines. In general, the sessions consisted of four AVG with an average duration of 8 min, and the multicomponent exercise was performed between the AVG. The multicomponent exercise lasted 13.1 min on average per session, divided into two or three activities with different objectives depending on the planning.

In the main part, the AVG included were the following: the Xbox 360^®^ with the Kinect using “Kinect Adventures” and “Kinect Sport”; the Nintendo Wii^®^ using “Wii Sports”, “Just Dance”, and “Mario and Sonic at the Olympic Games”; dance mats using “Dance Dance Revolution” and “Mario and Sonic at the Olympic Games” adapted from the Nintendo Wii to the dance mats; the BKOOL^®^ interactive cycling simulator connected to a tablet HUAWEI MediaPad T5 AGS2-W09; and the Nintendo Switch^®^ using “Ring Fit Adventures”. It is noteworthy that the Ring Fit Adventures is a novel AVG that has not yet been studied. The intervention was carried out in two locations, the University of Zaragoza and the “San Braulio” public school in Zaragoza. All the AVG were provided through funding, and each site was equipped with the AVG necessary to develop the intervention. The sessions were different every day, following a progression of difficulty and intensity and fulfilling the objectives previously established in the planning. The participants did not play all the AVG in each session, so the number of sessions recorded for each AVG is different. The order in which the activities were carried out was different among the participants, as each participant started in an AVG and changed it after playing.

The AVG were combined with multicomponent exercises focused on enhancing health-related physical fitness, such as cardiorespiratory fitness, muscular endurance, and/or muscular strength, but also coordination and balance. This intervention design combining AVG with traditional exercise was selected due to a potentially greater energy expenditure [30]. The multicomponent exercise performed had a playful background to enhance motivation and enjoyment.

### 2.3. Outcomes

#### 2.3.1. Physical Fitness

A walking-graded protocol was employed to assess cardiovascular fitness. Starting at a comfortable walking pace (3.2 km/h), speed was increased by 0.8 km/h every 2 min until the participants walked quickly (5.6 km/h), which was the maximum speed reached during the test. Then the slope was increased by 4% every minute until exhaustion (up to maximum of 20%). The test was performed on a treadmill (Quasar Med 4.0, h/p/cosmos, Nussdorf-Traunstein, Germany) with the mask fitted. A medical doctor, specialized in sports medicine, supervised the whole test. Respiratory gas exchange data were measured “breath-by-breath” using open circuit spirometry (Oxycon Pro, Jaeger/Viasys Healthcare, Hoechberg, Germany). Previously to maximal testing, an experienced physician examined each participant and gave permission to perform the cardiovascular fitness testing. The test was explained to the participants, who were fitted with electrodes and had their resting heart rate measured before starting.

Peak values of oxygen uptake and heart rate were defined as the highest average values obtained for any continuous 15 s period. The metabolic cart was calibrated with a gas of composition and volume previously known prior to the first test each day, as recommended by the manufacturer.

Electrocardiography (ECG) was used to record heart rate, utilizing a 12-lead system before and during the whole test. The maximal heart rate value was the highest value of heart rate recorded during the last stage of exercise. The blood pressure was also measured with a digital monitor (M3, HEM-72OO-E, Omron Healthcare Europe, Hoofddorp, Netherlands), for health and safety reasons, before the maximal effort test with the participant lying in a tilt, and during the recovery period in standing position, both on the right arm. The cuffs were adjusted to the circumference of the tested arm, and the measurement was taken twice. The participants had to be at rest 5 min before the pre-testing measurement.

#### 2.3.2. Accelerometer and Heart Rate Monitor

The children’s physical activity duration and intensity was objectively quantified using triaxial accelerometers (ActiGraph-GT3X + BT, ActigraphTM, LLC, Fort Walton Beach, FL, USA). The children wore the accelerometers during the whole session. These devices were placed on their left hip, which has been shown to be the region providing the most reliable data using this technique [37]. The accelerometers were used to record the heart rate measured by the Polar H10 (Polar Electro Oy., Lake Success, NY, USA) heart rate sensor and the Pro Chest Strap via Bluetooth. The children wore the Polar H10 heart rate sensor throughout the sessions, along with the accelerometer. It was important to ensure the correct placement of the chest strap and to avoid interference with other participants who were also wearing the band and the accelerometer.

#### 2.3.3. Energy Expenditure Estimate

A maximal effort test was performed by all the participants. In order to estimate the oxygen consumption during all parts of the session based on the registered heart rate, a linear regression equation was calculated for each participant based on the heart rate and oxygen consumption data during all stages of the treadmill test. The energy expenditure during each exercise and the whole session was then estimated using this equation developed for each child and the heart rate data obtained from the accelerometers and the Polar H10 heart rate sensor. This method is valid and reliable to measure the energy expenditure [38,39], and is even more reliable than the accelerometry data [40], also taking into account that one of the devices used consists of a cyclosimulator, which might not be reliably registered by accelerometry.

### 2.4. Statistical Analyses

The Statistical Package for the Social Sciences (SPSS) version 22.0 (SPSS Inc., Chicago, IL, USA) was used to perform all the statistical analyses. Statistical significance was set at *p* = 0.05 in all tests. Data are presented as mean and standard deviation (SD). Kolmogorov–Smirnov tests were performed to verify the normal distribution of the variables, and the variables that did not show a normal distribution were transformed. A one-way analysis of variance (ANOVA) was used to compare the energy expenditure of each AVG. Independent t tests were used to examine the gender and BMI status differences in the energy expenditure of each AVG. Cohen’s d effect sizes (95% confidence intervals (CI)) were calculated and interpreted as small (0.2–0.5), medium (0.5–0.8), or large (>0.8). A linear regression was performed for each participant with their individual oxygen consumption and heart rate data from the maximal test to calculate the energy expenditure when playing AVG.

## 3. Results

A total of 45 children and adolescents (10.1 ± 1.0 years) were included in this study. The participant characteristics are detailed in Table 1. Boys weighed more than girls, were taller, and had a higher BMI (all *p* < 0.05; Table 1).

The results showed 19 children with overweight (42.2%) and 26 children with obesity (57.8%) according to Cole et al. [36].

### 3.1. Energy Expenditure by Device

The energy spent in kcal/min and the metabolic equivalent (METs) for the whole session for each AVG and traditional exercise are reported in Table 2. The results showed an average energy expenditure of 315.1 ± 77.5 kilocalories for a one-hour session of this intervention, including AVG and exercises. Significant differences in energy expenditure were found between BKOOL and the Nintendo Wii (5.38 ± 1.2 vs. 4.38 ± 1.0 kcal/min; *p* < 0.05; Table 2). All the AVG included required a moderate intensity of between 3 and 6 METs [41].

### 3.2. Gender Differences

Significant differences in energy expenditure between boys and girls were found for the average of the sessions and for some AVG (Figure 1). Boys had a higher energy expenditure than girls when playing Ring Fit Adventures (5.90 ± 2.9 vs. 3.85 ± 1.2 kcal/min), Xbox Kinect (5.11 ± 0.9 vs. 4.01 ± 0.9 kcal/min), and BKOOL (5.85 ± 1.2 vs. 4.79 ± 0.9 kcal/min) (all *p* < 0.05; Figure 1a). These differences disappeared when using weight-independent metrics (METs), as shown in Figure 1b. This suggests that the differences shown in kcal/min were largely due to the difference in weight between boys and girls. Further analyzing the differences in the kcal/min spent between boys and girls, these differences in kcal/min disappeared when the VO_2max_ and fat-free mass mas were introduced as covariates, except for Xbox Kinect. No significant differences were found between boys and girls in the average heart rate recorded when playing each AVG.

### 3.3. BMI Status

No significant differences in energy expenditure and average of heart rate were observed between those with overweight and those with obesity, but a tendency was seen towards a higher energy expenditure in those with obesity in kcal/min (Figure 2a), which changed when the energy expenditure was expressed in METs (Figure 2b), indicating again an influence of weight on this energy expenditure.

## 4. Discussion

The aim of this study was to determine the energy expenditure of AVG sessions combined with exercise, as well as to analyze the energy expenditure of each device used and to investigate the differences between girls and boys. The results showed a mean energy expenditure of 315.1 ± 77.5 kilocalories in a one-hour session using different AVG combined with exercise. It should be noted that the energy expenditure of a complete session includes warming up (8–10 min per session), cooling down (5–8 min), and exercises (mean of 11min) other than the AVG. Regarding gender, some differences in energy expenditure were observed between boys and girls, being partially explained by differences in body weight. In addition, these differences in the energy expenditure expressed in kcal/min disappeared when VO_2max_ and fat-free mas were introduced as covariates, except for Xbox Kinect, in which there may be other factors that can influence energy expenditure, such as enjoyment and motivation, which were not measured in this study.

The energy expenditure in AVG was lower compared with participation in team sports (between 450 and 600 kcal/h), but it should be taken into account that obese children do not feel often ready, willing, or motivated to participate in any sport activity with their counterparts.

Children and adolescents spend much of their free time in sedentary activities, such as playing video games or watching TV (1–1.5 METs) [42]. Through AVG, the participants increased energy expenditure, but the key is to motivate sports participation at the end of the AVG intervention and, above all, to develop a healthier and more active lifestyle.

In accordance with the findings obtained in this study, previous studies that used the Nintendo Wii (Wii Sports or Just Dance), Xbox Kinect (Kinect Sports or Kinect Adventures), and Dance Dance Revolution reported similar results, with a higher energy expenditure in boys than in girls. Most of the previous studies that investigated AVG used an indirect calorimetry by using a portable metabolic analyzer to measure energy expenditure [43,44,45,46,47,48,49,50,51,52,53,54], but also a SenseWear^®^ armband was used [55,56,57]. Other studies monitored activity using accelerometers or recorded heart rate and estimated energy expenditure [58,59,60]. The estimation of energy expenditure through heart rate data, as was used in this study, is a valid, reliable, and practical method which is based on the linear relationship between heart rate and VO_2max_ [38,39]; indeed, this is the most reliable non-calorimetric method measuring energy expenditure at different intensities [40] without the need for a portable metabolic analyzer. One strength of the method is the individualization through the stress tests performed on the participants, which, according to Keytel et al. [38], is especially important.

Research using the Nintendo Wii is wide. The present study showed that the Nintendo Wii was the device that required less kilocalories, with an energy expenditure of 4.38 ± 1.0 kcal/min. These findings are in line with a number of previous studies; Graf et al. [43] showed energy expenditure values of 2.2–2.86 METs and 3.86–3.9 METs in children when playing Wii Sports (bowling and boxing, respectively). Significant differences were found between boys and girls in energy expenditure (METs) for Dance Dance Revolution (level 1) and Nintendo Wii bowling, but not in Dance Dance Revolution (level 2) and Nintendo Wii boxing. Another study performed by Lau et al. [47] reported that Chinese children spent 2.05–5.14 kcal/min when playing Nintendo Wii, and no significant differences were shown between normal weight and overweight or obese children, concluding that AVG could provide an opportunity to increase physical activity, regardless of BMI status. A comparison of the energy cost of playing two games of Nintendo Wii between children with healthy weight and overweight or obesity was performed by O’Donovan [48]. In this study, AVG play resulted in being of light to moderate intensity, which contributes to an increase in daily energy expenditure, but this does not seem to be enough. Again, no differences were seen between normal weight and overweight or obese children, supporting the use of video games specially to treat childhood obesity.

Graves et al. [49] investigated the energy expenditure during three different games of Wii Sports, and showed that boxing produced the highest exergy expenditure (4.05 kcal/min), followed by tennis (3.05 kcal/min) and bowling (2.75 kcal/min), reporting less energy expenditure than the present study. Similar results were found by Lanningham-Foster et al. [44], who showed an increased energy expenditure of 3.15 kcal/min in children by playing Nintendo Wii Boxing, higher than in adults (2.47 kcal/min), and also reporting a lower energy expenditure than the current study. Another study investigated the energy spent by young boys playing several AVG of the Nintendo Wii, establishing Wii Sports boxing (3.05 METs) as the AVG that spent the most energy, followed by Wii Step (2.43 METs), Wii Sports tennis (2.16 METs), Wii Sports bowling (2.03 METs), and Wii Sports ski (1.65 METs) [50]. This study classified the Nintendo Wii as a low-intensity activity and compared this intensity with the intensity of walking at 4.5 km/h. This intensity is lower than the intensity reported in the present study (4.6 ± 1.2 METs). Siegmund et al. [51] reported energy expenditure values of 12.17 mL·kg·min for Wii Sports boxing (1.93 kcal/min), a much lower energy expenditure than the one reported in the current study. As shown, the studies discussed above show a lower energy expenditure compared to the present study. This may be due to an overestimated energy expenditure based on the results of the peak stress test and the heart rate recorded during game time, as the Nintendo Wii mainly requires movement of the upper extremities. Thus, playing the Nintendo Wii will be an activity with less muscle recruitment and therefore less energy expenditure. Another explanation could be that the participants of the present study were constantly supervised and encouraged during the sessions with AVG.

Quan et al. [58] measured 28 AVG sessions with Wii Fit, Just Dance, and Wii Sports to determinate how much time the participants spent at different intensities. The results showed that the participants spent 19.9% in moderate to vigorous physical activity, 32.9% in light physical activity, and 47.2% was sedentary time; no significant differences by gender were observed for moderate to vigorous physical activity, light physical activity, and sedentary behavior. In relation to the physical activity levels, one study compared the children’s physical activity levels in physical education, play time, and the Nintendo Wii [59]. The results showed that children had the highest amount of moderate to vigorous and light physical activity, and the lowest sedentary time by playing Nintendo Wii, compared with physical education and play time. The highest sedentary time part was for physical education. Therefore, AVG could increase light and moderate to vigorous physical activity among children.

Several other articles investigated the potential of Wii Sports and Dance Dance Revolution to increase physical activity, concluding that these AVG elicit positive effects on energy balance [43,52]. It is important to note that, in the present study, dance mats were used not only to play Dance Dance Revolution, but were also used to play Mario and Sonic at the Olympic Games, adapted from the Nintendo Wii, which increased the energy expenditure. According to Graf et al. [43], the energy expenditure during Dance Dance Revolution and Wii Sports boxing were comparable to moderate-intensity walking (at 5.7 km/h), but during Wii Sports bowling it was less intense. The energy expenditure reported were lower compared with the present study, possibly due again to the method used to estimate energy expenditure. It is noteworthy that Mario and Sonic at the Olympic Games was adapted from the Nintendo Wii to the dance mats and was used to play at dance mats, recording the energy expenditure along with Dance Dance Revolution, which can increase the energy expenditure of the dance mats. Significant differences were found between girls and boys by Graf et al. [43], with a 19–33% higher energy expenditure (kcal/h per weight (kg)) for boys when playing DDR level 1, DDR level 2, and bowling, and similarly the VO_2_ was 20% to 34% higher in boys (*p* < 0.05) during DDR level 1, DDR level 2, and bowling. With these data, it could be checked if, when including the maximum oxygen consumption and the fat-free mass as covariates, these differences still exist or disappear, as in the present study. Similar results were shown by Bailey et al. [52], who displayed an energy cost of 4.2 METs for Wii Sports boxing and 5.5 METs for Dance Dance Revolution, similar to the results shown in this study. It should be noted that the mean age was higher in the study by Bailey et al. [52] compared to the present study (11.5 ± 2.0 vs. 10.1 ± 1.0 years respectively). Bailey et al. [52] found no differences by BMI status for energy expenditure, and children with overweight or obesity had a higher enjoyment. Furthermore, a systematic review that included articles related to energy cost for Nintendo Wii, Eye Toy, and Dance Dance Revolution reported that the physical activity intensity of these AVG is light to moderate but that the energy expenditure was significantly lower for AVG played primarily through upper limb movements compared to those that engaged the lower limbs [61]. Nintendo Wii is primarily played with the upper extremities, which would explain why both the results of the current study and previous scientific evidence report a lower energy expenditure. Furthermore, motivation and enjoyment seem to influence the energy expenditure when playing the Nintendo Wii, as the amount and speed of movement will determine that expenditures. Nonetheless, although the contribution to the energy expenditure of the Nintendo Wii is less than that of other AVG, it is well known that some activity is better than none [48]. The Nintendo Wii could be a good tool to help increase the daily energy expenditure of overweight or obese children who are reluctant to engage in physical activity.

Xbox Kinect is another device often used in AVG interventions. The energy expenditure by Xbox Kinect has been studied, and it seems to elicit a higher intensity than Nintendo Wii, probably because it requires movement of the whole body. The previous scientific evidence agrees with the results obtained in this study. Most studies that evaluated energy expenditure playing Xbox 360 Kinect reported significant differences between girls and boys in kilocalories spent, as this study did. Smallwood et al. [53] showed the energy expenditure of two games of Xbox Kinect (Dance Central and Kinect Sports boxing) in school children between 11 and 15 years old. The energy expenditure was 172 kcal/h. Specifically, for Kinect Sport boxing (one of the AVG used in this study), the energy expenditure was 4.4 METs. Gender differences were found when playing Kinect Sport Boxing for the VO_2_ and energy expenditure, with boys showing a significantly higher energy cost than girls (5.1 kcal/min for the boys compared with 3.4 kcal/min for the girls), and this difference remained significant (*p* = 0.004) when the caloric expenditure was normalized for body weight. This supports the results obtained in the present study. Similar results were found by Clevenger et al. [54], who reported an energy cost of 4.6 METs from 335 AVG sessions using the Xbox Kinect, and some AVG such as Kinect Adventures and Wipe Out exceeded 6 METs. Comparisons by groups were performed, showing that males spent significantly more kilocalories per minute than females, which supports the findings of this study; overweight or obese participants had a greater energy expenditure that healthy-weight participants, and teens spent more energy than children, which can be explained by the difference in weight. Two more studies found significant differences in the energy expenditure between males and females [45,60]. Vallabhajosula et al. [60] compared the energy expenditure in Xbox Kinect sessions using “Reflex Ridge” from Kinect Adventures vs. the energy expenditure during regular play time. The results showed higher kilocalories per min and METs in an Xbox Kinect session (1.85 kcal/min and 4.21 METs) in comparison with regular play time (1.59 kcal/min and 3.7 METs), and a higher percentage of very vigorous activity was seen for theXbox Kinect condition. No significant interactions were observed for BMI or gender, but there was a trend toward significance, where the rate of energy expenditure achieved was 34% higher in males. The energy cost in two AVG from Xbox Kinect (“River Rush” and “Reflex Ridge”) was measured by McNarry et al. [45], showing that those AVG elicited moderate-intensity physical activity (5.5–5.7 METs), and could be used for meeting physical activity daily recommendations. The results also reported significant differences between males and females in one of the AVG played. As stated above, the energy expenditure playing AVG is similar to the energy spent during a brisk walk. Canabrava et al. [55] agree with this comparison as their results showed the similarity in energy expenditure between walking at 5 km/h and playing Xbox Kinect (Kinect Adventures, Kinect Sports boxing, and a dance videogame), reporting an energy cost of 185 kilocalories per hour and showing that AVG can be an interesting alternative to increase physical activity and to replace traditional sedentary video games. Once again, there were no significant differences in energy cost between children with a normal weight and children with overweight or obesity.

Some articles have shown differences by player mode in Xbox Kinect, comparing single vs. multiplayer mode. Barkman et al. [46] reported significant differences in energy expenditure between the single-player mode, with an energy cost of 15.4 mL·kg·min, and the multiplayer mode, with an energy cost of 16.8 mL·kg·min. METs for several AVG were also reported, with an energy cost of 3.9, 3.8, 3.5, and 3.1 METs for the single-player mode and 4.1, 4.3, 3.7, and 3.2 METs for the multi-player mode of Wipe Out, Kinect Adventures, Kinect Sports, and Just Dance, respectively. In accordance with these results, Verhoeven et al. [56] studied the same conditions, concluding that children consumed more energy in multi-player mode, as reported by Barkam et al. [46]. In addition to the mode of play, another factor that can affect energy expenditure is the narrative of the AVG. According to Sousa et al. [62], the narratives in AVG were associated with moderate to vigorous physical activity and with higher average heart rate, without increasing the rate of perceived exertion.

It might have been interesting to have studied the differences in energy expenditure when comparing the sample of overweight or obese children with a group of normal-weight children. Hwuang et al. [63] studied the differences in energy expenditure among children with normal weight and children with overweight or obesity. The results showed that children in both groups expended similar energy relative to their weight, and AVG were able to elicit moderate to vigorous intensity physical activity for all children, potentially contributing to meet the recommended physical activity levels. However, overweight/obese children spent more time at light intensity, and children with normal weight engaged more in vigorous-intensity activity than those with overweight/obesity.

Some studies go further, such as the study performed by Gao et al. [59], which proposed to integrate AVG within school curricula, with positive results as it contributed to children’s daily energy expenditure and moderate to vigorous and light physical activity as much as physical education did, and even with a higher intensity.

AVG interventions need to be structured and supervised to produce positive effects on energy expenditure. No effects were found on physical activity or energy expenditure in home-based and/or unstructured AVG interventions [34,35]. In the present study, not only was a planned and structured intervention carried out, but it was combined with traditional physical exercise with the aim of increasing energy expenditure. This increased the total expenditure of the sessions. In addition, the intervention combined different AVG to achieve motivation based on novelty and variety and to adapt to the preferences of the participants. In this way, the AVG intervention performed in this study goes beyond other AVG interventions. In addition, novel AVG such as BKOOL and Ring Fit Adventures were included in the intervention of the present study, which have hardly been studied.

There is less scientific evidence about AVG interventions using a bicycle simulator or an interactive stationary bike. The BKOOL could be considered a game-based exercise strategy rather than an active video game, used primarily with adults. However, it could be a tool to promote active exercise among children and adolescents because it is an interactive and novel tool. Adamo et al. [64] investigated the effects of an interactive AVG with a stationary bike on overweight or obese adolescents, showing an average energy expenditure of 576.2 kcal/h. Another study performed an AVG intervention with stationary bikes which consisted of cycling to control a virtual tank. The energy expenditure with this AVG was 7.6 kcal/min in young adults [57]. In both articles, the energy spent by AVG on stationary bikes was higher than the expenditure obtained with BKOOL in this study, but the participants of those articles were young adults.

Finally, the Ring Fit Adventures is a recently released AVG, which has not yet been studied to the best of our knowledge. Ring Fit Adventures includes cardiorespiratory fitness and strength exercises, as well as dynamic and static stretches. During the AVG, there are “fitness battles” where strength exercises are used as attacks to defeat virtual enemies. These strength exercises are focused on the legs (such as squats, thigh presses, or knee lifts), arms and chest (such as bow pulls, triceps kickbacks, chest presses, and overhead presses), and core (such as planks, standing twist, and leg raises from the seated or lying positions). There are even yoga positions such as the chair, warrior, and tree poses to train balance and strength. The player has to skip to move through the game, raising their knees further to climb stairs or pass across the water, which is a demanding cardiorespiratory training. Ring Fit, together with BKOOL, are the AVG that required the most energy expenditure in kcal/min in this study, probably because they required full-body movement.

A higher energy expenditure of BKOOL and Ring Fit Adventures and a lower energy expenditure when playing Nintendo Wii can be explained by the muscles involved. Related to these results, a meta-analysis performed by Peng et al. [29] reported that AVG that mainly involved upper-limb movements had significantly smaller effects on energy expenditure than AVG that mainly involved lower-limb or whole-body movements.

Reviews of AVG show positive effects on energy balance and physical activity, helping to decrease sedentary behaviors [29,30,31]. It is still unclear if the intensity achieved during AVG is enough to help children meet the recommended physical activity levels [32,33], but the results of the present study are in line with those shown by previous scientific evidence, that at least moderate intensity is achieved with AVG. In this study, a wide variety of AVG were measured to find out the energy expenditure of the same group of children with overweight or obesity playing different devices. The results showed that the BKOOL was the device that notably required the most energy, and the Nintendo Wii the one that demanded the least. The energy expenditure when playing AVG was between 4.63 and 5.61 METs. In addition, differences in the kilocalories per minute were reported between boys and girls, which were mostly due to the weight difference. The main novelty was the fact of applying a combined intervention of AVG and exercise to achieve a higher energy expenditure. Another difference with previous research was the inclusion of the Ring Fit from the Nintendo Switch and the BKOOL cycling simulator into the AVG intervention. At the moment, no article has been published including Ring Fit, so this article is the first to report energy expenditure data for this new AVG.

Future research may focus on comparing interventions with AVG and other forms of training and on investigating how to make the most of this promising tool.

## 5. Conclusions

In view of our results, a structured and supervised AVG intervention combined with multi-component exercise is an effective strategy to produce moderate-intensity physical activity and to increase the energy expenditure in children and adolescents with overweight and obesity. Ring Fit Adventures is a promising tool to increase energy expenditure, exceeding the energy expenditure of the other AVG except BKOOL. The energy expenditure seems to be higher for boys in comparison with girls, but these differences might be partially due to disparities in body weight. In conclusion, AVG are an innovative and interesting tool that can help fight against childhood obesity and its future consequences.

## 6. Limitations and Strengths

Some limitations should be addressed. The overall energy expenditure for each device was measured, and no measurements were reported for each game used in each AVG. Reporting the kilocalories spent for each game used in each AVG would have been interesting in order to know which of them was the most energy consuming for each device. The energy expenditure was measured with the heart rate data through the maximum stress test data, since a portable metabolic analyzer was not available. In addition, the game mode was not taken into account, and evidence shows that using single-player or multi-player modes affects the energy expenditure, but most of the sessions were performed in pairs. Due to the characteristics of the intervention, it was complicated to record motivation or enjoyment during the AVG, but it would have been interesting to see the relationship between these two variables and energy expenditure.

On the other hand, this study is notable for the wide variety of AVG investigated. In addition, Ring Fit Adventures and BKOOL were included in the sessions with AVG, two novel devices that offer opportunities and possibilities to significantly increase energy expenditure in overweight or obese children. The BKOOL stands out for its interactivity, and the Ring Fit stands out for its playability and its focus on physical exercise and fitness. Despite not having portable metabolic analyzers, the individualized calculation of the energy expenditure through the data of the stress test and the heart rate measured during the sessions was a reliable and effective method [38,40,65], even in children [65]. Thanks to this method, it was possible to measure the energy expenditure of a large number of sessions and participants. Finally, the AVG intervention in this study was combined with multi-component exercise, which allowed us not only to increase the total energy expenditure of the sessions but also to compare the expenditure in the different AVG with the expenditure during exercise.

## Figures and Tables

**Figure 1 ijerph-17-06714-f001:**
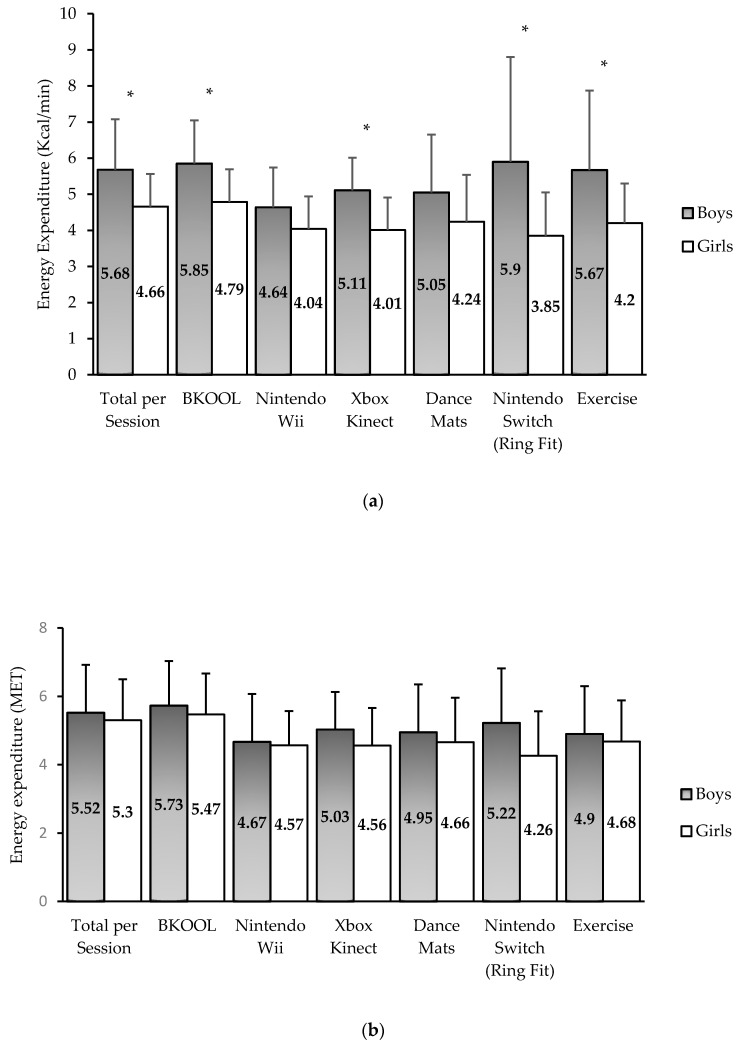
Energy expenditure for the whole session, each AVG, and exercise without AVG categorized by gender. * Indicates a statistically significant difference between boys and girls (*p* < 0.05). (**a**) Energy expenditure measured in kcal/min; (**b**) Energy expenditure measured in MET.

**Figure 2 ijerph-17-06714-f002:**
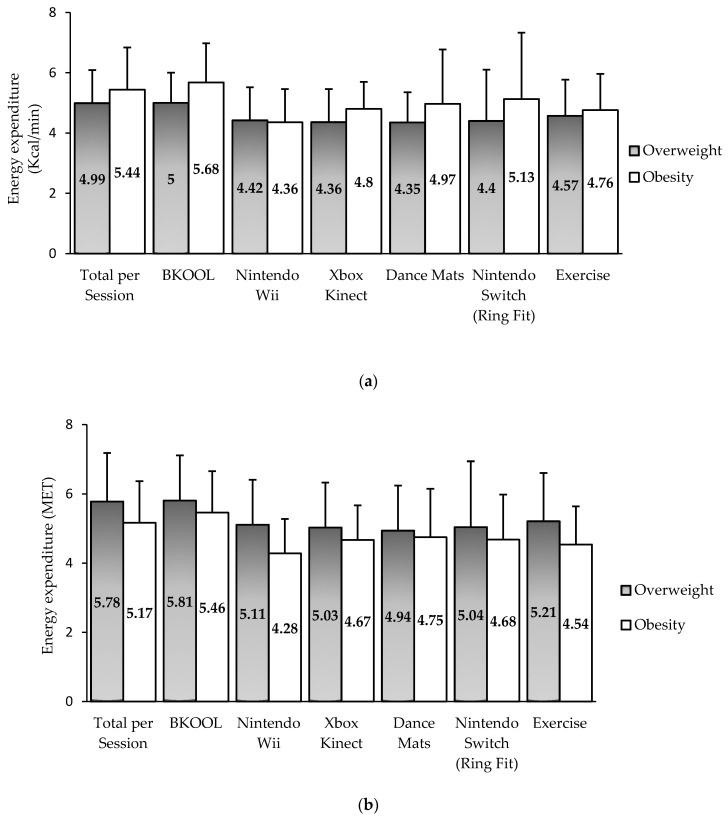
Energy expenditure for the whole session, each AVG, and exercise without AVG categorized by BMI status. (**a**) Energy expenditure measured in kcal/min; (**b**) Energy expenditure measured in MET.

**Table 1 ijerph-17-06714-t001:** Participant characteristics.

**Divided by Gender**					
**Variable**	**All (N = 45)** **Mean ± SD**	**Boys (*n* = 26)** **Mean ± SD**	**Girls (*n* = 19)** **Mean ± SD**	***d***	***p*** **Value**
Age (years)	10.1 ± 1.0	10.3 ± 1.0	9.8 ± 1.0	0.20	0.116
Weight (kg)	56.3 ± 10.7	60.1 ± 10.9	51.0 ± 8.1 *	0.36	0.004
Height (cm)	146.6 ± 7.6	149.1 ± 6.0	143.1 ± 8.4 *	0.37	0.008
BMI ^a^ (kg/m^2^)	26.0 ± 3.2	26.9 ± 3.6	24.8 ± 2.3 *	0.26	0.030
BMI percentile	97.22 ± 2.4	97.6 ± 2.1	96.7 ± 2.7	0.39	0.201
VO_2peak_ (mL·kg·min)	32.5 ± 5.4	32.8 ± 6.1	32.1 ± 4.4	0.06	0.650
**Divided by BMI status**					
		**Overweight (*n* = 19)** **Mean ± SD**	**Obesity (*n* = 26)** **Mean ± SD**	***d***	***p*** **Value**
Age (years)	-	10.6 ± 0.7	10.1 ± 1.0	0.49	0.117
Weight (kg)	-	49.9 ± 6.9	60.9 ± 10.7 ^†^	1.22	0.000
Height (cm)	-	145.5 ± 7.2	147.3 ± 8.0	0.24	0.438
BMI ^a^ (kg/m_2_)	-	23.4 ± 1.5	27.9 ± 2.9 ^†^	1.94	0.000
BMI percentile	-	95.4 ± 2.7	98.58 ± 0.5	1.65	0.000
VO_2peak_ (mL·kg·min)	-	34.0 ± 5.3	31.4 ± 5.2	0.50	0.106

^a^ BMI: Body Mass Index; * indicates a statistically significant difference between boys and girls (*p* < 0.05); ^†^ indicates a statistically significant difference between children with overweight and obesity (*p* < 0.05); Cohen’s *d* can be small (0.2–0.5), medium (0.5–0.8), or large (>0.8).

**Table 2 ijerph-17-06714-t002:** Energy expenditure for the whole session, each AVG ^a^, and exercise without AVG ^a^.

Variable	N	Kcal/min (*n* = 45)	METs ^a^	Average Heart Rate
Complete session	45	5.25 ± 1.3	5.42 ± 1.3	134.52 ± 14.4
BKOOL	43	5.38 ± 1.2	5.61 ± 1.2	145.7 ± 12.0
Nintendo Wii	40	4.38 ± 1.0 *	4.63 ± 1.2 *	133.76 ± 10.9 *
Xbox Kinect	44	4.64 ± 1.0	4.83 ± 1.1	135.20 ± 13.5 *
Dance Mats	40	4.70 ± 1.5	4.83 ± 1.3	136.59 ± 12.2 *
Nintendo Switch (Ring Fit)	23	4.88 ± 2.0	4.80 ± 1.5	137.57 ± 14.8
Exercise	44	4.68 ± 1.2	4.81 ± 1.3	132.94 ± 12.2 *

^a^ AVG: active video game; METs: metabolic equivalents; * indicates a statistically significant difference with BKOOL (*p* < 0.05).

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
