# Peer review of "Assessment of Active Video Games’ Energy Expenditure in Children with Overweight and Obesity and Differences by Gender"

_ijerph, 2020, doi:10.3390/ijerph17186714_

Round 1

Reviewer 1 Report

The work presented to me for review is very interesting, it raises a new, interesting and developmental topic. However, I have a few questions:

  1. What were the exlusion criteria?
  2. The authors used many AVG tools and many games. Did each study participant used each device? Did the authors of the study provide the equipment, or did the study only involve people who had AVG at home?
  3. How were the blood pressure and heart rate measured? Were you referred to percentile grids?
  4. On which hip was the accelerometer? What parameters were taken into account and according to what algorithms? What criteria had to be met for the accelerometer data to be considered valid and correct?
  5. In table 1 it is worth adding the p values ​​for each parameter
  6. In table 2 - again, has each participant practiced on each type of console or e.g. 5 people on Xbox, 3 people on Wii, etc.
  7. In my opinion, it would be useful to have a control group of children with a healthy body weight to check what energy expenditure will be like for them.

Reviewer 2 Report

The article entitled “Assessment of active video games’ energy expenditure in children with overweight and obesity and differences by gender” aimed to investigate the energy expenditure during an AVG intervention combined with exercise and compare by sex. The study brings some interesting results using new technology. However, some issues need to be addressed in their introduction and discussion. Additionally, a major methodological issue may or may not change the results/conclusions. Please see specific comments.

1- Abstract and Objective: To “determine” the energy expenditure may give the impression of the determination of new cut-off points to be used as a new standard for the given population. Given the non-representative size of the sample, the authors objective was to “investigate” the EE.

2- Nintendo Switch (Ring Fit) is cutting-edge technology in AVG. The investigation of any kind in the new device should be highlighted in the introduction and objectives. Especially what makes this new AVG different from different consoles.

3- Please provide information on how the participants were assessed for Tanner stages.

4- Children should be clustered by BMI percentile, rather than BMI. Please make all proper adjustments in methods, results, discussion, and conclusion in that regard. See: https://www.cdc.gov/obesity/childhood/defining.html

Barlow SE and the Expert Committee. Expert committee recommendations regarding the prevention, assessment, and treatment of child and adolescent overweight and obesity: summary report. Pediatrics 2007;120 Supplement December 2007:S164—S192

Whitlock EP, Williams SB, Gold R, Smith PR, Shipman SA. Screening and interventions for childhood overweight: a summary of evidence for the US Preventive Services Task Force. Pediatrics. 2010;116(1):e125—144external icon.

5- Add p-value for each comparison in Table 1.

6- If the participants were in the same maturational stage, how do the authors explain differences in height and weight? Could this interfere with the results?

7- BMI percentile should be included in Table 1.

8- Table 2 should provide more info regarding each “activity” such as average time, average heart rate, PA data (from accelerometers). These data should also be compared statistically between the “activities”.

9- Additional movement and physiological data should also be reported and compared between weight status and sex, such as step count, time spent in MVPA, and average HR.

10- Back to methods, is not clear whether participants underwent the activities in the same order (within- and between participants).

11- Although AVGs seem to be a feasible method to promote moderate PA, research shows that children do not play it for a sufficient amount of time. Additional methods such as narratives are being introduced and tested to increase time and engagement of gameplay.

Robinson TN, Banda JA, Hale L, Lu AS, Fleming-Milici F, Calvert SL, et al. Screen media exposure and obesity in children and adolescents. Pediatrics 2017 Nov;140(Suppl 2):S97-101 [FREE Full text] [doi: 10.1542/peds.2016-1758K] [Medline: 29093041]

Sousa CV, Fernandez A, Hwang J, Lu AS. The Effect of Narrative on Physical Activity via Immersion During Active Video Game Play in Children: Mediation Analysis. J Med Internet Res 2020;22(5):e20134. URL: https://www.jmir.org/2020/5/e20134. DOI: 10.2196/20134

12- Some rationale to compare EE by sex needs to be provided in the introduction.

13- Additional rationale and discussion regarding the intervention needs to be addressed. For example, what is the ecological validity of the protocol? People do really reproduce something similar to the protocol in real-life?

Round 2

Reviewer 1 Report

Please describe the blood pressure measurement methodology. It is very important.
- in what position was tested
- on which arm was the cuff
- whether the cuffs were adjusted to the circumference of the tested arm
- how much the person had to be at rest before the measurement 
- how many times the measurement was taken

Author Response

Some information has been included about the blood pressure measurement methodology in line 189: "The blood pressure was also measured with a digital monitor (M3, HEM-72OO-E, Omron Healthcare Europe, Hoofddorp, Netherlands), for health and safety reasons, before the maximal effort test with the participant lying in a tilt and during the recovery period in standing position, both on the right arm. The cuffs were adjusted to the circumference of the tested arm and the measurement was taken twice. The participants had to be at rest 5 minutes before the pre-testing measurement."

Reviewer 2 Report

No further comments. 

Author Response

Thank you for your comments.